# Numerical Modeling of Non-Uniformly Reinforced Carbon Concrete Lightweight Ceiling Elements

**Rostislav Chudoba [1],\***, **Ehsan Sharei [2]**, **Tilo Senckpiel-Peters [3] and Frank Schladitz [3]**

[1]  RWTH Aachen University, 52062 Aachen, Germany
[2]  H+P Ingenieure GmbH, 52072 Aachen, Germany; esharei@huping.de
[3]  Teschnische Universität Dresden, 01069 Dresden, Germany; tilo.senckpiel-peters@tu-dresden.de (T.S.-P.);
    frank.schladitz@tu-dresden.de (F.S.)
\*  Correspondence: rostislav.chudoba@rwth-aachen.de; Tel.: +49-241-8028150



**Featured Application:** The present paper contributes to the discussion on modeling methods appropriate for the structural analysis of thin-walled concrete shells, a rapidly developing field of material and structural design utilizing the high-performance cementitious composites reinforced with non-metallic reinforcement. An effective modeling support is paramount for the derivation of reliable and economic design and assessment principles in a wide range of applications.

**Abstract:** The paper focuses on the specifics of macro-scale modeling of thin-walled textile-reinforced concrete shells. Application of layered shell finite elements requires systematic procedures for identification of material characteristics associated with the individual layers within the cross section. The identification of the material parameters describing the tensile behavior of a composite cross section is done using data obtained from the tensile test. Such test is usually performed only for a reference configurations with a simple layup of fabrics and a chosen thickness. The question is how to derive the strain-hardening response from the tensile test that is relevant for a changed cross-sectional configuration. We describe and discuss scaling and mixture rules that can be used to modify the material parameters for modified cross-sectional layups. The rules are examined in the context of the test results obtained on a shell that was reinforced non-uniformly, with varying types of textile fabrics and varying thickness within the shell surface.

**Keywords:** textile-reinforced concrete; thin-walled shells; cementitious composites; layered finite elements; mixture rules; model calibration

## 1. Introduction

Several applications of thin-walled concrete shells reinforced with high-performance textile fabrics realized in the recent decade have convincingly demonstrated the potential of the new type of composite for the design and construction of highly efficient structural members [1,2]. The combination of fine aggregate concrete matrix with textile fabric reinforcements made of carbon enabled the construction of lightweight, thin concrete shells with curved geometries [3]. Experimental investigations of structural behavior were performed for large-scale shells with strain-hardening behavior serving as roof elements [4–6], pedestrian bridges [7,8], sandwich panels and facade elements [9–11]. Because of the non-corrosive nature of the reinforcement, a high sustainability and durability of this type of structures are provided. The development of compatible material components, i.e., the cementitious matrix and the reinforcement fabrics, is continuously extending the spectrum of design and manufacturing options. This development of material and manufacturing

technologies calls for an effective support in terms of efficient and validated modeling approaches delivering a correct prediction of the structural behavior of thin-walled textile-reinforced concrete (TRC) shells. Efficient and realistic models of non-uniformly reinforced fabric reinforced shells are the assumption for the development of robust design and assessment concepts [12,13]. Such modeling strategies must account for the specific features of the material and of the nonlinear structural behavior of TRC shells. The particular issues to be considered in the macro-scale modeling of TRC shells include:

- Correct reproduction of the strain-hardening response of the shell cross section for a wide range of loading conditions including the tensile or bending loads.
- Initial and damage induced anisotropy owing to the oriented crack pattern within the two-dimensional stress state in the shell plane.
- Geometrical nonlinearity in interaction with imperfections requiring the buckling analysis as documented in experimental studies [4] and numerical investigations of the effect of imperfections on the structural behavior [14,15].

The focus of the present paper is on the first item, i.e., a flexible reflection of the strain-hardening behavior in macro-scale simulations of TRC shells. The second aspect of damage-induced anisotropy is considered here as well; however, it does not have a significant influence on the structural behavior of the studied girder. The issue of geometrical nonlinearity and buckling behavior is not within the scope of the present paper.

Considering a TRC element with sufficient reinforcement ratio loaded in tension, three phases of tensile response are distinguished:

- Phase I: linear elastic behavior with the strain–stress relation characterized by the composite. elasticity modulus,
- Phase II: formation of the cracks at the level of the matrix tensile strength,
- Phase III: saturated crack state with steady matrix stress level between the cracks; the effective stiffness of the composite in this phase is approximately equal to the stiffness of the reinforcing fabrics.

Existing approaches to modeling of this tensile behavior can be roughly classified into three categories summarized in Figure 1. The theoretical descriptions based on discrete representation of the matrix cracking process and debonding between matrix and reinforcement displayed in column (a) of Figure 1 considers an elastic-brittle behavior of fabrics and matrix material components. In this model, the nonlinear strain-hardening response results from the evolution of discrete cracks emerging along the specimen with random matrix strength and nonliner bond stress—slip relation [16,17]. The ambition of this modeling approach is to predict the strain-hardening behavior of a particular cross-sectional design using the parameters of the material components and of the bond between the concrete and fabrics [18].

Even though the discrete crack modeling approaches provide a valuable insight into the process of matrix cracking and debonding, a realistic and reliable prediction of the composite response $\sigma_c(\varepsilon_c)$ is still not possible. The reason is the complexity of interaction mechanisms governing the bond behavior. Moreover, the effect of material heterogeneity and imperfections at the levels of the fiber bundles, of the fabrics and of the cross-sectional thickness introduces further challenging tasks for the development of micro- and meso-scale modeling approaches. At this level of modeling, further effort is necessary to capture the inelastic interaction effects within a multi-scale model with a broader range of validity than available so far.

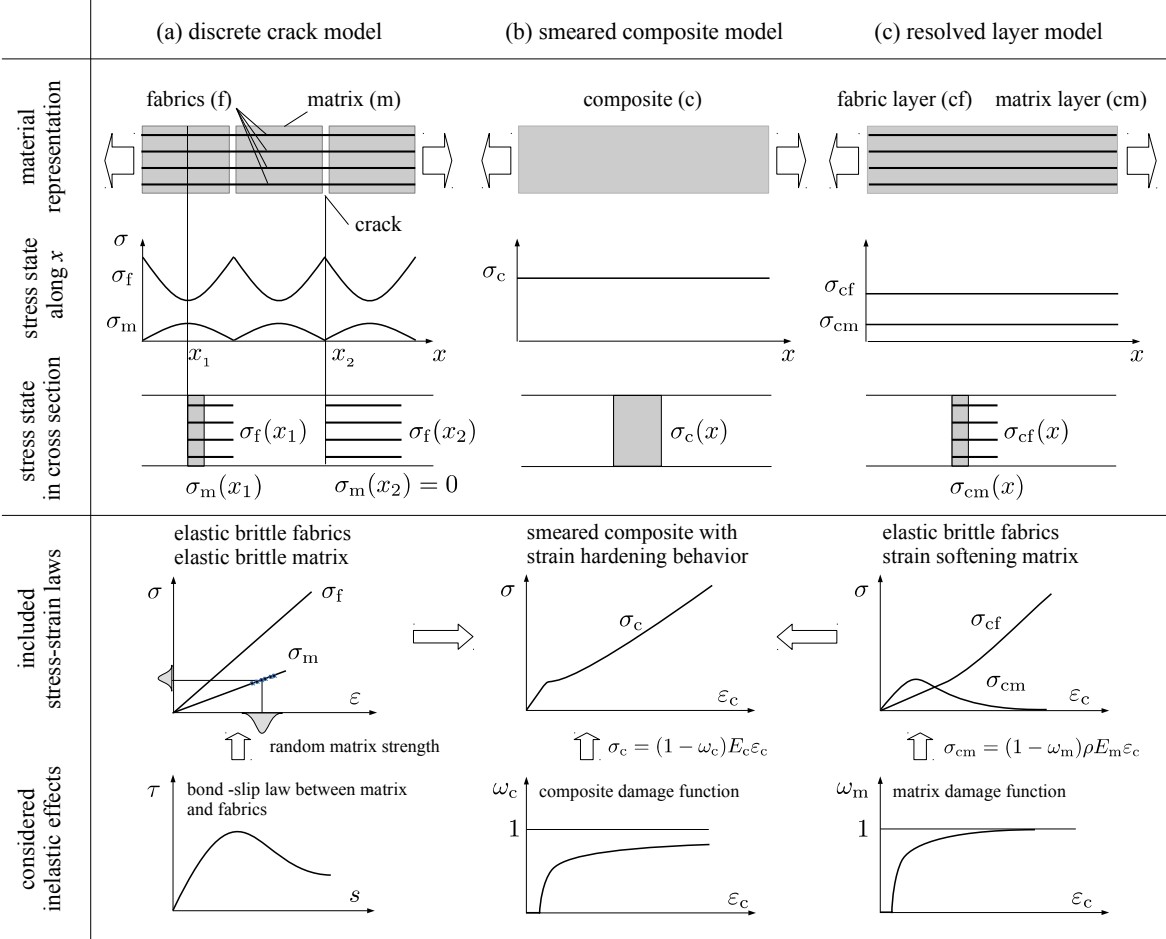

**Figure 1.** Classification of modeling approaches to tensile behavior of textile-reinforced concrete shells including the resolution of stresses along a uniaxial specimen, resolution of stress and strain components in a material point and examples of inelastic material characteristics.

As a result, the strain-hardening response of the composite is usually experimentally characterized using a composite tensile test. The measured response provides the input for an efficient calibration of phenomenological numerical material models. These macro-scale material models, applicable in finite element simulations, represent the strain-hardening response phenomenologically in terms of inelastic state variables, i.e., damage and plastic strains. An approach representing the composite with a uniform material behavior over the cross section depicted in column (b) of Figure 1 has been applied by the authors for the simulation of with laminated carbon TRC shells [14]. In this simulation concept, the material parameters are specific to a particular cross-sectional layup and thickness. If the cross-sectional configuration changes, the material model is not valid anymore and its material parameters must be recalibrated. Due to the plane stress state within the shell surface, the inelastic effects, i.e., damage or yielding, need to be captured only in the in-plane direction of the shell. The application of this model is limited to cross sections with uniform fabric reinforcement over their height or to structures with prevailing membrane stresses. As discussed later on, the reason is that the position of the fabrics within the cross section is not distinguished in the model so that the bending behavior of sparsely reinforced cross sections cannot be correctly reproduced.

In this paper, we focus on the modeling approach depicted in column (c) of Figure 1 with a resolved representation of the matrix and fabric layers within the cross section. In contrast to the discrete cracking models, the matrix cracking represented in a smeared way along the specimen length but is ascribed to a resolved layer of concrete within the shell cross section. Such a distinction between the effective fabrics and matrix layers allows for the rescaling of material parameters for a changed

cross-sectional configuration including e.g., reinforcement ratio or layup of the fabrics within a cross section. Such scaling is needed to enhance the validity range of the macroscopic models for layered TRC shells so that a smaller amount of calibration experiments is required for the identification of the material parameters. The model classification in Figure 1 is focused on the representation of a the composite cross section. Finite element models based primarily on solid finite element discretization with an explicit resolution of material interfaces including also the localization of individual concrete cracks, presented e.g., in [19,20], have a different focus and different purpose, i.e., local stress effects in structural details of the simulation of TRC sandwich panels. For an efficient and realistic prediction of the nonlinear behavior of thin TRC shells, the layered shell models provide the appropriate dimensional reduction of the simulated boundary-value problem.

Layered shells have already been applied recently in the context of the nonuniformly reinforced TRC shells using both smeared and resolved layup within a TRC shell [21,22] in combination with damage-plasticity models available in *ABAQUS* [23]. Consistent with these studies, a generalized description of the scaling and mixture rules for elastic and inelastic material parameters reflecting modified layups of thin TRC shells is proposed in the following sections. The smeared and resolved models of the cross section are validated using the test data obtained within an experimental investigation of girder element briefly described in Section 2. The smeared representation of a composite cross section (Figure 1, column b) is realized using an anisotropic damage model characterized in Section 3. The decomposition of the tensile response for the layered cross section (Figure 1, column c) is then described in detail in Section 4 including the qualitative validation of the correct bending response. Finally, in Section 5, we present the results of the numerical simulations of the girder response performed using the calibrated material models in combination with the smeared and resolved cross-sectional idealizations.

## 2. Test Setup for the Carbon Concrete Girder

The experimental response of the thin-walled carbon concrete ceiling elements depicted in Figure 2 developed at the Institute of Concrete Structures at TU Dresden [24] served for the validation of the macroscopic modeling approaches (c) and (d) from Figure 1. The girders were non-uniformly reinforced with two types of fabrics of the fineness 3300 tex and 800 tex. Their thickness was non-uniform as well. The cross-sectional shape is shown in Figure 3a. As apparent from Figure 3b, two types of cementitious matrix were applied within the cross section, the outer layers were made of the Fleece Concrete Composite (FCC) and the inner textile-reinforced concrete layer (denoted as Carbon Reinforced Concrete in Figure 4) was made of a fine grain concrete with the carbon fabrics position in the middle of the 10 mm high section. The geometry, layout of fabrics within the shell and the specification of layers within a cross section of the girder is provided in Figure 4.

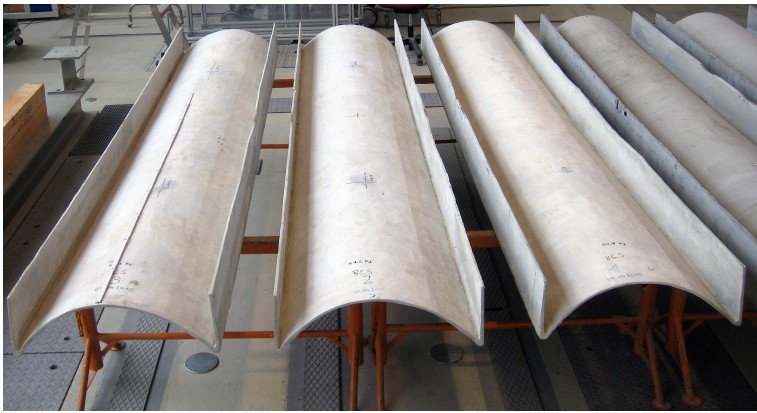

**Figure 2.** Test specimens.

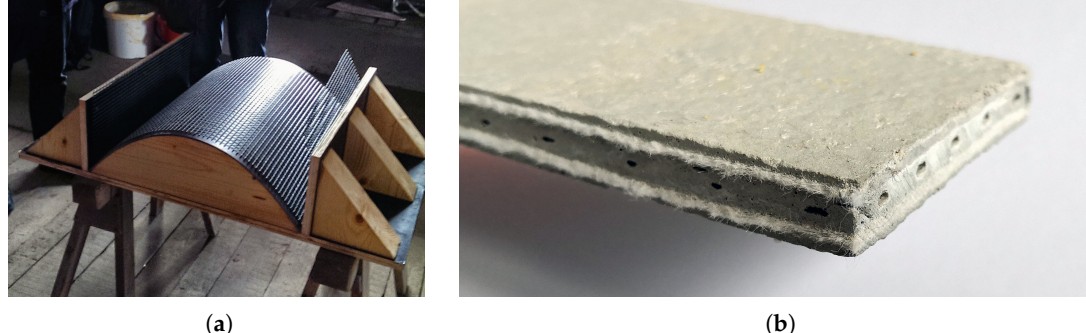

**Figure 3.** Girder cross section: (**a**) profile of the girder with the carbon fabric reinforcement; (**b**) cross-sectional layup consisting of carbon reinforced concrete in the middle section and fleece concrete composite at the top and at the bottom.

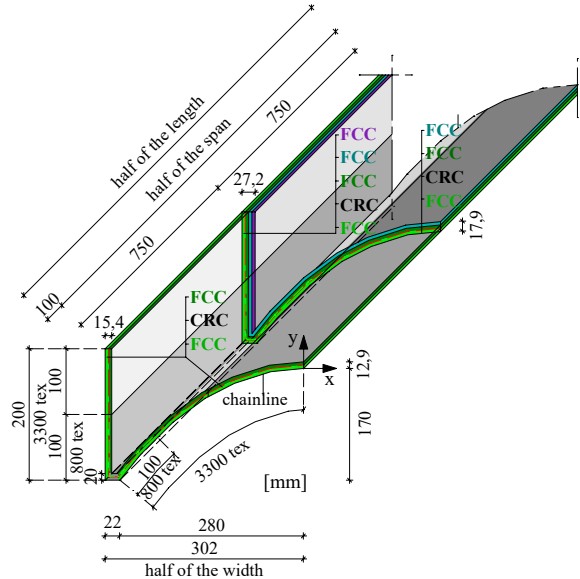

**Figure 4.** Cross section layups of reinforced shell sections consisting of 3300 tex and 800 tex fabrics, fleece concrete composite (FCC) and carbon reinforced concrete (CRC) displayed for a quarter section of the girder corresponding to the two planes of symmetry [24].

The girders were tested using a six-point bending test depicted in Figure 5. The load was introduced using wooden plates following the catenary curve of the vault within a girder section. The experimentally obtained load–deflection curves obtained in the test are used later on in Section 4 for the validation of the discussed modeling approaches.

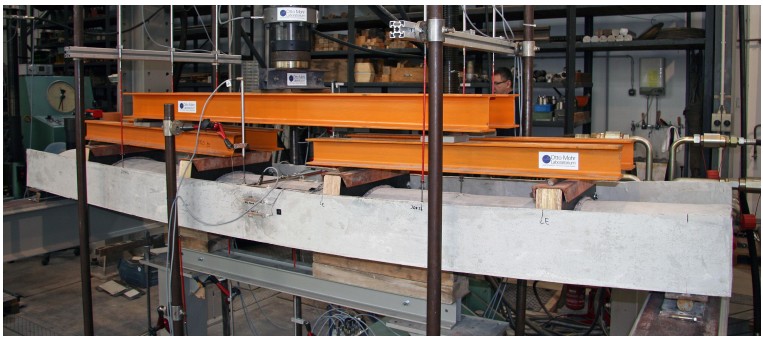

**Figure 5.** Test setup of the carbon concrete girder.

## 3. Smeared Model of a Cross Section

### 3.1. Characterization of the Applied Anisotropic Damage Model of a Shell Layer

The classification of modeling approaches introduced in Figure 1 considers the inelastic material behavior of the composite layers in a general way. Even though a damage model is chosen in the last row of Figure 1 as an example, the classification can be used in combination with plasticity models describing the strain-hardening or strain-softening behavior as well. This classification provides the framework for the explanation of the scaling and mixture rules for inelastic material parameters presented in the sequel.

To formulate and to validate the scaling and mixture rules in combination with a particular example of an inelastic material model, a microplane damage model introduced in [25] has been chosen to reflect the nonlinear material behavior of the composite layers. Inelastic effects governing the stress–strain response are thus represented by a damage function as exemplified in the last row of Figure 1. The key idea behind the microplane models is to reflect the material state on a set of planes on a unit hemisphere or circle around a material point instead of using the usual tensorial representation. The microplanes are used to establish the mapping between the strain and stress tensors as proposed in the original model by Bažant et al. [26] sketched in Figure 6a. The macroscopic strain tensor $\varepsilon$ is decomposed onto the microplane directions by a geometric projection, delivering the microplane strain vectors $e$. A constitutive law at the microplane level defines the relation between the apparent and effective microplane strain and stress vectors. Finally, the macroscopic stress tensor $\sigma$ is obtained using the condition of energetic equivalence between the microplane discretization and the tensorial representation of stress at the material point. As the original version was detected to be thermodynamically inconsistent, several refinements have been proposed as documented e.g., in [27]. Currently, the microplane discretization of a material state provides a well established framework for sound formulation of anisotropic inelastic material models [28].

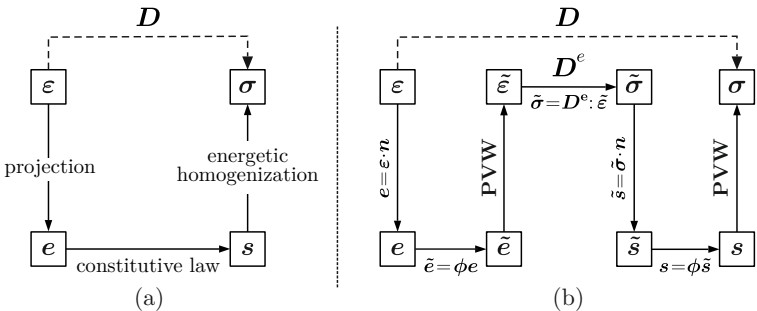

**Figure 6.** Algebraic structure of the microplane models (**a**) basic principle of microplane model; (**b**) constitutive stress–strain relation in the microplane damage model.

In the microplane damage model used here [25], each microplane is associated with a projected strain and damage state variables. Its important property is the separation between the apparent and effective stresses and strains. As indicated in the algebraic structure of the microplane damage model depicted in Figure 6b, the apparent stress tensor $\varepsilon$ is first projected onto the microplanes rendering the microplane strain vectors $e$. Then, the effective microplane strains $\tilde{e}$ is obtained with the help of the damage/integrity function. By employing the condition of energetic equivalence, i.e., the principle of virtual work, the corresponding effective strain tensor $\tilde{\varepsilon}$ is obtained by integrating the strains over the microplane discretization. The relation between the effective strain tensor $\tilde{\varepsilon}$ and the effective stress tensor $\tilde{\sigma}$ is provided explicitly using the elasticity tensor $\boldsymbol{D}^e$. Analogous to the described mapping of strain, the mapping between the effective stress tensor $\tilde{\sigma}$ and apparent stress tensor $\sigma$ is performed with the help of the microplane state discretization governed by the prescribed integrity function $\phi = 1 - \omega$. In the applied implementation of the model, only the normal component of the microplane

strain and stress is considered. The model is thermodynamically consistent and implicitly captures the Poisson's effect of the undamaged, still an effective skeleton of a material structure in a material point.

The briefly characterized microplane damage model falls into the category of anisotropic damage models. Its application to the simulation of TRC shells has been described in [14]. The reason for choosing this model in this paper is twofold: the damage function can be automatically calibrated for both strain-hardening and strain-softening behavior, and it can reflect the anisotropy of damage due to the development of oriented cracks in the in-plane directions of the shell. The applied version of the model specialized for shells uses microplanes arranged around a unit circle centered at a material point. By aligning the stress and strain tensors in a material point to the shell geometry, the in-plane $(\xi, \eta)$ and out-of-plane $(\zeta)$, the evolution of in-plane damage at a material point can be related to a polar discretization of the material state in a material point displayed in Figure 7. In this state representation, the degree of damage depends on the orientation within the shell surface. This fact introduces the damage anisotropy reflecting the evolution of oriented, fine crack pattern.

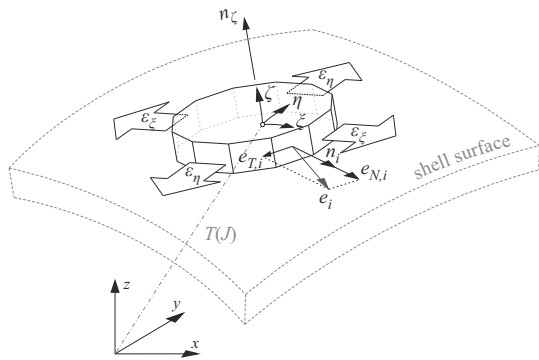

**Figure 7.** Microplane model within a single layer of a shell cross section.

Further details of the mathematical formulation of the microplane damage model for TRC shells would go beyond the scope of the present paper. The complete model description including the calibration procedure, elementary verification studies and model validation using three-point bending test and slab tests are provided in [14]. The formulated material model was implemented by the first two authors as a user subroutine in the finite element code *ABAQUS* and in the in-house research software *BMCS*. It is used in the following studies both in smeared and in resolved versions of the shell model to simulate the strain-hardening and strain-softening material behavior, respectively.

*3.2. Calibration of a Smeared Cross Section Model Using a Tensile Test*

To identify the damage functions with the same effective stress–strain behavior assumed in all material points of a cross section, the data from the tensile test following the RILEM recommendation [29] was used as input. The cross sections with similar fabric layups as those used in the tested girder were tested in two series. The specimens were reinforced with a single layer of fabrics, one with 3300 tex and one with 800 tex. Figure 8 shows the stress–strain response obtained in four tests for each cross section as gray dashed lines, and the average response curves as black solid lines. Given the cross-sectional thickness $d^{\text{test}}$ and area of the textile fabrics per unit width $a_{\text{f}}$, the reinforcement ratio $\rho^{\text{test}}$ reads

$$\rho^{\text{test}} = \frac{a_{\text{f}}}{d^{\text{test}}}. \tag{1}$$

The fabric area per unit length and the corresponding reinforcement ratios for the tested specimens reinforced with both types of fabrics are listed in Table 1.

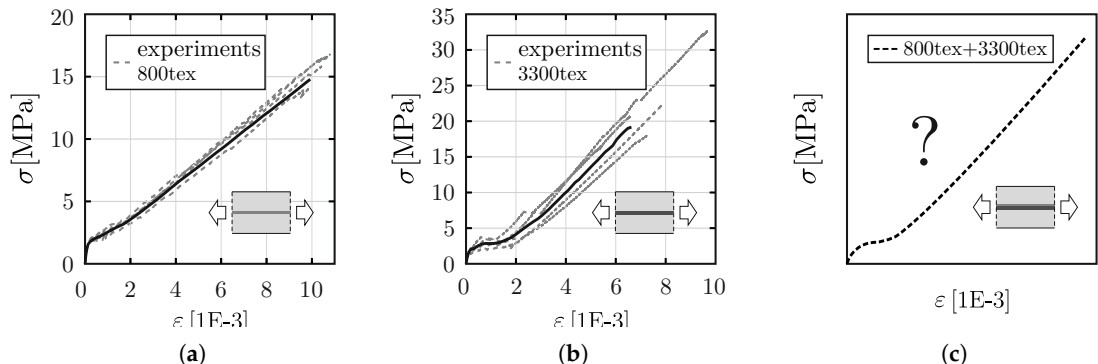

**Figure 8.** Tensile response of the cross sections with (**a**) 800 tex, (**b**) 3300+tex reinforcement and (**c**) of the hybrid cross section 3300+800 tex.

**Table 1.** Cross section characteristics; parameters of performed tests in boldface font, derived values in normal font.

| | | | 800 | 3300 | 3300 + 800 |
|---|---|---|---|---|---|
| fabric fineness | | [tex] | **800** | **3300** | 3300 + 800 |
| fabric area | $a_f$ | [m$^2$/m] | **0.616 · 10$^{-4}$** | **1.713 · 10$^{-4}$** | 2.329 · 10$^{-4}$ |
| specimen thickness | $d$ | [m] | **0.01** | **0.01** | 0.01 |
| reinforcement ratio | $\rho = a_f/\bar{d}$ | [-] | 0.616 · 10$^{-2}$ | 1.713 · 10$^{-2}$ | 2.329 · 10$^{-2}$ |
| hybrid section ratio | $\eta$ (Equation (9)) | [-] | 0.264 | 0.735 | 1.000 |
| final composite stiffness | $E_{cf}$ | [GPa] | 1.48 | 3.22 | 4.70 |
| effective fabric stiffness | $E_f$ | [GPa] | 250 | 188 | 202 |

The parameters of the microplane damage model include the Young's modulus ($E_m = 28$ GPa) and Poisson's ratio (0.2) and the function $\phi(e) = 1 - \omega(e)$ defining the diminishing integrity evolution for an increasing normal microplane strain. The integrity functions obtained using the incremental calibration procedure are plotted in Figure 9a,b for the two tested cross sections. As observed for the cross section with 800 tex fabrics in Figure 9a, the integrity does not drop to zero but remains at a constant level corresponding to the final elastic stage of the strain-hardening behavior at which no further matrix cracks appear. In case of the 3300 tex specimens, the shape of the damage function shown in Figure 9b reveals an increasing integrity in the range of strains $0.001 < \varepsilon < 0.002$. This effect is owing to the delayed activation of filaments within the yarn cross section, usually referred to as slack, of stretched filaments in this range of strains.

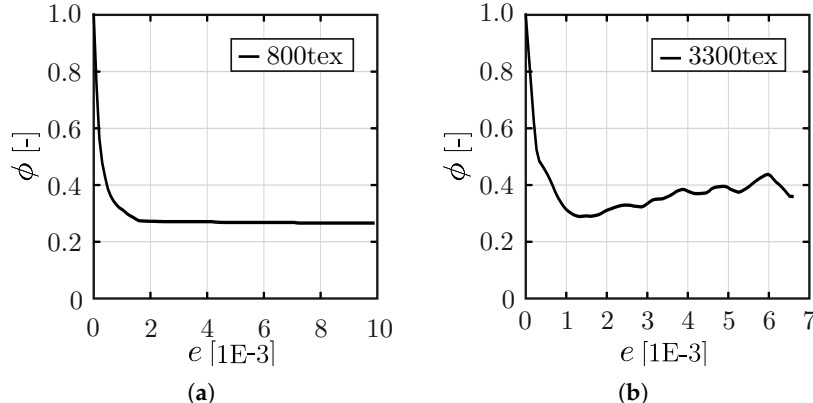

**Figure 9.** Calibration of integrity functions $\phi = (1 - \omega)$ using experimentally obtained stress–strain curves on 10 mm thick specimens reinforced with one layer of (**a**) 800 tex and (**b**) 3300 tex carbon fabrics.

### 3.3. Limitations of the Smeared Cross Section Model

The calibrated functions are valid only for the cross sections that were used in the test. Any change in the cross-sectional configuration would require conducting a new tensile test with a subsequent calibration of inelastic parameters. Such a requirement would certainly make the design, calibration, structural analysis and assessment of shells uneconomic. Thus, scaling procedures for strain-hardening response are required to extend the validity range of numerical models for a wider range of cross-sectional designs.

Another limitation of the smeared model results from the assumption of uniform material behavior in all layers of the cross section. Figure 10 shows the strain and stress profiles over the cross-sectional height that occurs upon bending load. Since every material point in the cross section follows the same strain-hardening curve, there is no possibility to reflect the effect of the position of the fabrics in the cross section. As a result, the smeared model cannot correctly capture the effect of the lever arm on the bending response in sparsely reinforced cross sections. As we shall document later on, this fact can result in an overestimation of stiffness if the loading induces non-negligible amount of bending stresses within an analyzed shell.

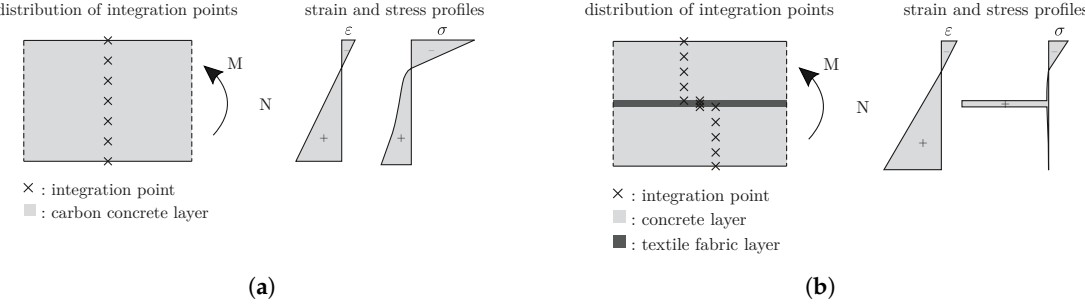

**Figure 10.** Stress and stress and profiles for the considered representations of the composite cross section with the distributions of integration points over the height: (**a**) smeared idealization with uniform material in each material point; (**b**) resolved cross-sectional idealization with different material behavior in plain concrete and fabric layers

## 4. Resolved Model of a Cross Section

To extend the range of applications to shells with a non-uniform distribution of fabrics within the cross section, the following enhancements of a material model describing the layer behavior are required:

- Scaling of the strain-hardening response for a modified reinforcement ratio related to a varying shell thickness or to the area of fabrics.
- Mixture rule to identify the strain-hardening response of a cross section layer combining several types of fabrics.
- Definition of a layered discretization of a shell cross section based on the scaling and mixture rules.

As already mentioned, these enhancements are not specific to the underlying microplane damage but are also valid for other types of inelastic material models, e.g., damage-plasticity models.

### 4.1. Decomposition of the Composite Stress

To derive the constitutive parameters related to a resolved, layered representation of a cross section within a shell finite element, a simple idealization of the cross section consisting of two parallel uniform material components with nonlinear, behavior is used. By idealizing the cross section as two nonlinear, springs with a stiffness defined in accordance with area fractions of the matrix and fabrics, the stress–strain measured in the reference tests are decomposed into matrix and fabrics stresses as follows:

$$\sigma_c^{\text{test}} = \sigma_{cf}^{\text{test}} + \sigma_{cm}^{\text{test}} = \rho^{\text{test}} \sigma_f + (1 - \rho^{\text{test}}) \sigma_m, \tag{2}$$

where $\sigma_{cf}^{\text{test}}$ and $\sigma_{cm}^{\text{test}}$ are related to the unit area of the composite and $\sigma_f$ and $\sigma_m$ to the area of the material of fabrics and matrix, respectively.

Since the stress–strain response of the studied carbon fabric material is linear elastic and brittle, a natural choice for the approximation of the fabric stress is $\sigma_f = E_f^{\text{yarn}} \varepsilon_c$, leading to the fraction of composite stress transmitted by fabrics

$$\sigma_{cf}^{\text{test}}(\varepsilon_c) = \rho^{\text{test}} E_f^{\text{yarn}} \varepsilon_c, \tag{3}$$

with $E_f^{\text{yarn}}$ determined in a yarn tensile test.

However, the effective behavior of the fabrics in the composite may be significantly different from the one measured in the tensile test of a yarn. The major two sources of the difference are

- the nonuniform fabric strain along the composite tensile test specimen, and
- the nonuniform strain profile within a thick fiber bundle cross section.

The former effect of matrix fragmentation leading to a fluctuating stress transfer to and from the concrete matrix between the cracks indicated in column (a) of Figure 1 can be explained and visualized using the meso-scale models explicitly reflecting the multiple-cracking and debonding during the loading process [18].

The latter issue considers the non-uniformity of the stress within the yarn with roughly 50,000 filaments within the bundle as is the case for the 3300 tex carbon yarns used within the cross section. Even though these yarns are penetrated with styrene-butadiene material, their effective stiffness is much lower than in case of the 800 tex fabrics [30].

Moreover, another structural effect within the bundle that makes the transformation of material characteristics non-trivial can be recognized in Figure 11b. Fabrics with a large yarn cross-sectional area containing up to 50,000 filaments can exhibit a phenomenon called slack, meaning that the reinforcement reaches its full stiffness only after some initial level of loading. This effect occurs when a non-negligible amount of filaments is not aligned with the direction of loading so that they must be stretched before they can start to contribute to the load transfer.

The interaction of the inelastic effects, i.e., slack, multiple-cracking and debonding, explain why the material stiffness $E_f^{\text{yarn}}$ determined in yarn or fabric tensile tests is different from the effective stiffness in the composite. Therefore, a pragmatic approach to the identification of the effective fabric stiffness sketched in Figure 11 is used as a starting point in the decomposition of the composite stress.

This approach exploits the fact that, in the saturated state of cracking, the contribution of the matrix segments between the cracks to the composite stiffness is negligible, i.e., $E_c \approx E_{cf}$. Thus, the effective fabric stiffness $E_f$ can be directly determined from the final branch stress–strain curve as documented in Figure 11a for the test with 800 tex fabrics. Then, the effective stiffness of fabrics $E_f^{test}$ in the composite is evaluated as

$$E_f^{test} = E_{cf}^{test}/\rho^{test}.$$

In case of 800 tex fabrics, the effective stiffness $E_{cf}^{800} = 250$ GPa approaches a level measured in a yarn tensile test. On the other hand, in the case of the 3300 tex fabrics, the value $E_{cf}^{800} = 188$ GPa reveals that a large fraction of filaments within the cross section was not activated within the tensile test—see Table 1.

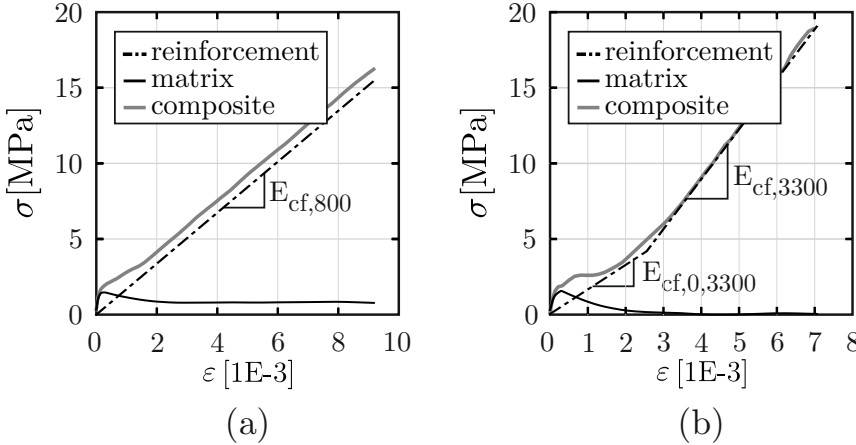

**Figure 11.** Stress–strain curves of the matrix (black solid lines), reinforcement (dashed lines) and composite (grey solid lines) for cross sections with (**a**) 800 tex and (**b**) 3300 tex reinforcements.

The effect of slack observed for the tests with 3300 tex fabrics has been treated by assuming a bilinear approximation of the fabric stress $\sigma_{cf}(\varepsilon_c)$ within the composite as shown in Figure 11b.

Using the obtained approximations of the effective fabric stress $\sigma_{cf}^{test}$, the corresponding fraction of matrix stress during the whole loading history is obtained as the difference between the composite and fabric stresses:

$$\sigma_{cm}^{test}(\varepsilon_c) = \sigma_c^{test}(\varepsilon_c) - \sigma_{cf}^{test}(\varepsilon_c). \tag{4}$$

The matrix stresses shown in Figure 11 grow up to the peak stress at the level of the first crack and start to diminish. The shape of the curves is similar to the stress–strain curve obtained for plain concrete in a three-point bending test. However, its mechanical interpretation is different. It does not reflect the strain-softening in the fracture process zone of a localizing macroscopic crack. It rather represents a smeared process of multiple cracking along a zone of a tensile specimen. In the former case, a stable crack growth with damage localization to a macroscopic crack is considered, while, in the latter case, multiple cracks emerge suddenly across a whole cross section in an unstable way.

### 4.2. Mixture Rule for Hybrid Fabric Reinforcement

In the girder member described in Section 2, four regions were reinforced with both types of fabrics i.e., 3300 + 800 tex. However, no tensile test that could be used for calibration of such a cross section has been conducted. Thus, to derive the stress–strain curve of such a cross section, a mixture rule must be used to extract the theoretical tensile behavior from the separately tested 3300 tex and 800 text cross sections.

Consider a hybrid cross section combining several types of fabrics $j = 1 \dots m$ with a cross-sectional area per unit length $a_f^{(j)}$. For each of these fabrics, a composite tensile test has been conducted

delivering the stress–strain curves $\sigma_c^{(j)}$. To combine the behavior of the fabrics in a hybrid cross section, the composite stress of each test $j$ must first be decomposed using the procedure described in Section 4.1 into the fractions associated with fabrics and matrix $(\sigma_f^{(j)}, \sigma_m^{(j)})$. Then, the fabrics stresses $\sigma_f^{(j)}$ can be mixed into a single hybrid reinforcement layer using the area fractions of each reinforcement type in the total reinforcement area $\bar{a}_f = \sum_j^m a_f^{(j)}$ with the weight factors

$$\eta^{(j)} = \frac{a_f^{(j)}}{\bar{a}_f}. \tag{5}$$

The effective fabric stress within the hybrid fabric cross section is then given as

$$\bar{\sigma}_f = \sum_i^m \eta^{(j)} \sigma_f^{(j)}. \tag{6}$$

The matrix stress is obtained by averaging the contributions determined in the individual tests

$$\bar{\sigma}_m = \frac{1}{m} \sum_i^m \sigma_m^{(j)}. \tag{7}$$

Then, the composite stress in a cross section of thickness $\bar{d}$ reinforced with a hybrid fabric reads

$$\bar{\sigma}_c = (1 - \bar{\rho}) \, \bar{\sigma}_m + \bar{\rho} \, \bar{\sigma}_f \tag{8}$$

with the reinforcement ratio

$$\bar{\rho} = \bar{a}_f / \bar{d}.$$

In reality, the usage of hybrid fabrics can introduce additional damage effects that are not reflected by this simple scaling of one-dimensional springs. In particular, a finer grid structure of overlapping fabrics reduces the contact area between the lower and upper concrete layers and can thus lead to surface delamination in the fabric plane at a low level of loading. However, in the case of the reinforcement ratio applied in the studied girder, the validity of the scaling can be assumed.

To relate the mixture rule to the studied girder, let us rewrite Equations (5)–(8) considering the case of hybrid reinforcement layer consisting of 3300 + 800 tex fabrics. Using cross-sectional characteristics of the test specimens summarized in boldface font in Table 1, the weighting factors of the fabric mixture rule read

$$\eta^{800} = \frac{a_f^{800}}{\bar{a}_f}, \quad \eta^{3300} = \frac{a_f^{3300}}{\bar{a}_f}, \quad \bar{a}_f = a_f^{800} + a_f^{3300}. \tag{9}$$

The fractions of fabric stress and of the matrix stress within a unit cross section are expressed as

$$\sigma_f^{3300+800} = \eta^{800} \sigma_f^{800} + \eta^{3300} \sigma_f^{3300}, \tag{10}$$

$$\sigma_m^{3300+800} = \frac{1}{2}(\sigma_m^{3300} + \sigma_m^{800}) \tag{11}$$

and the corresponding stress–strain curve of the composite cross section reads

$$\bar{\sigma}_c^{3300+800} = \left(1 - \rho^{3300+800}\right) \sigma_m^{3300+800} \tag{12}$$
$$+ \rho^{3300+800} \sigma_f^{3300+800}.$$

The result of this mixture rule is plotted in Figure 12 both as the composite stress fraction ascribed to the fabrics $\sigma_{cf}$ (a) and the composite stress $\sigma_c$ including also the matrix stress fraction $\sigma_{cm}$ (b). To provide a comparison of effective stiffness measured in the tests with 3300 tex and 800 tex fabrics,

the effective stiffness of the hybrid carbon textile fabric reinforcement in the final branch of the stress–strain curve is quantified in Table 1.

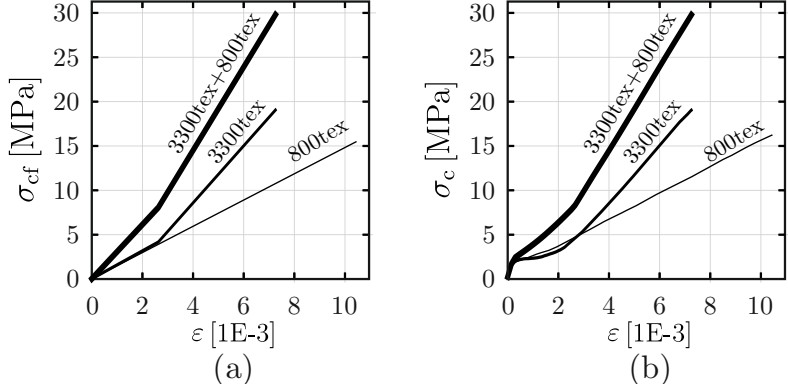

**Figure 12.** Reinforcement and composite stress–strain curves for cross sections with (**a**) single 800 tex and (**b**) 3300 tex reinforcements and the combined cross section.

### 4.3. Scaling of Composite Response for a Layer of a Shell Element

To identify the amount of stress corresponding to a layer of a finite element shell with a given layer thickness and its reinforcement ratio, the scaling formula can be used:

$$\sigma_{\rm c}^{(i)} = \frac{1 - \rho^{(i)}}{1 - \rho^{\rm test}}\sigma_{\rm cm}^{\rm test} + \frac{\rho^{(i)}}{\rho^{\rm test}}\sigma_{\rm cf}^{\rm test}. \tag{13}$$

This formula uses the decomposed stresses $\sigma_{\rm cm}^{\rm test}$ and $\sigma_{\rm cf}^{\rm test}$ evaluated from the composite stress–strain curve valid for the reinforcement ratio $\rho^{\rm test}$. The source stress–strain curve can be either the original curve measured in the test or it can be a result of the mixture rule described in Section 4.2. The decomposition into the matrix and fabric stresses is performed using the procedure described in Section 4.1.

To verify this scaling formula, let us consider the limiting cases of a composite layer with the reinforcement ratios 1 and 0 and substitute it into Equation (13) to obtain the stress in a layer that represents either the reinforcement or matrix materials, respectively as

$$\rho^{(1)} = 1 \implies \sigma_{\rm c}^{(1)} = \frac{\sigma_{\rm cf}^{\rm test}}{\rho^{\rm test}} = \sigma_{\rm f}, \tag{14}$$

$$\rho^{(2)} = 0 \implies \sigma_{\rm c}^{(2)} = \frac{\sigma_{\rm cm}^{\rm test}}{1 - \rho^{\rm test}} = \sigma_{\rm m}.$$

Thus, a layer corresponding to either plain matrix or plain fabric material is correctly recovered. By composing two layers such as these into a single cross section with the reinforcement ratio $\rho^{\rm test}$, the original composite stress obtained in the test can be recovered using Equation (2).

### 4.4. Calibration for Resolved Cross-Sectional Idealization

The last step needed for the characterization of an arbitrary composition of layered cross section consistently reproducing the tensile strain-hardening response is the calibration of the material model describing the effective strain-softening behavior of the concrete matrix layers. This nonlinear relation $\sigma_{\rm cm}(\varepsilon_{\rm c})$ is shown for the tested specimens in Figure 11. To reflect this behavior in the applied microplane damage model, the calibration of integrity functions $\phi$ representing the inelastic material parameters has been performed for the tested cross sections. The results are displayed in Figure 13.

As previously specified in (7), the contribution of the matrix stress $\bar{\sigma}_\mathrm{m}$ in a mixed cross section is introduced as an average of contributions $\sigma_\mathrm{m}^{(j)}$ determined from the tensile tests characterizing the matrix behavior in combination with the applied types of fabric reinforcement $j$. With regard to this fact, the contribution of the matrix stress to the cross section is nearly proportional to the area of matrix within the cross section. Thus, when composing a layup in a shell finite element with varying thickness, it is possible to simply add an effective matrix layer of a corresponding thickness and material behavior $\bar{\sigma}_\mathrm{m}(\varepsilon_\mathrm{c})$ without the need to recalibrate the material parameters of the layers in the cross section.

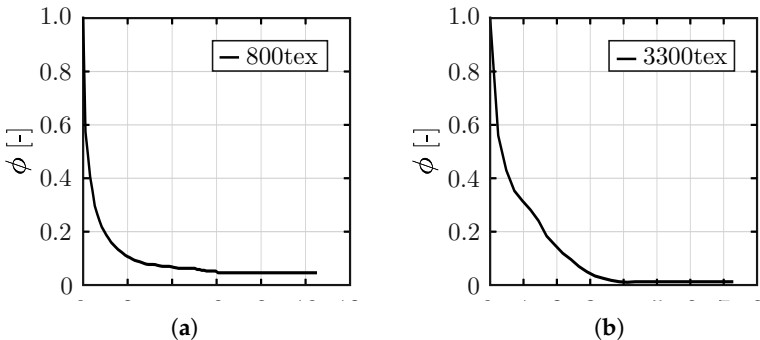

(a) (b)

**Figure 13.** Damage functions describing the matrix cracking calibrated using the matrix stress–strain curve in Figure 11 extracted from the composite tensile tests with (**a**) 800 tex and (**b**) 3300 tex.

*4.5. Verification of the Resolved Approach for Bending*

As mentioned in Section 3.3, a correct prediction of the bending response is one of the requirements that leads to the usage of resolved, layered representation of the shell cross section. To test the ability of the resolved model to correctly represent the development of an effective lever arm within a cross section, a parametric study of a three-point bending test with the span of 500 mm and width of 100 mm was performed with three different positions of the fabric within the cross sections. A discrete load was applied at the mid-span of the beam. The cross section with the height of 10 mm was reinforced with one layer of carbon fabric with 3300 tex. The behavior of the fabrics was assumed isotropic within the reinforcement layer with the stress–strain curve $\sigma_\mathrm{f}^{3300+800}(\varepsilon_\mathrm{c})$ and a layer thickness adjusted such that it matches the area of the fabric $a_\mathrm{f}^{3300+800}$ within the cross section. The studied configurations (i), (ii) and (iii) are shown in Figure 14 together with the corresponding load–deflection curves obtained from the simulation. The deflection was recorded using the midspan section of the girder.

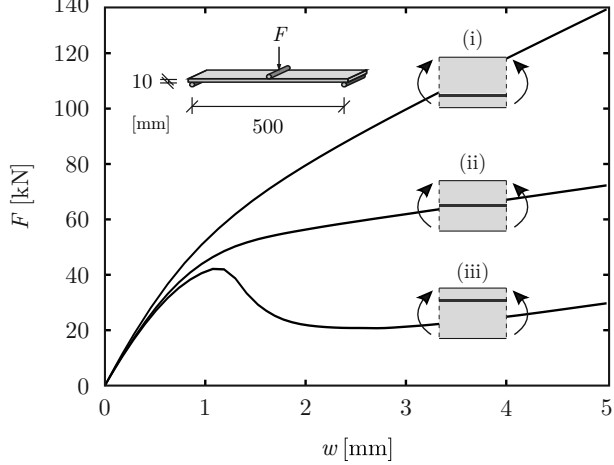

**Figure 14.** Load displacement curves of the simulation of carbon concrete beam with variation of the reinforcement position in the cross section subjected to bending action.

As apparent from the figure, the effect of the placement of the fabric is reproduced plausibly. The placement of the fabric at the bottom of the cross section in the tensile zone leads to a large lever arm and, thus, to an activation of a higher amount of compressive stresses within the cross section. As a result, this study shows the structural response with high stiffness and ductility (iii). The qualitative effect of reduced lever arm for configurations (i) and (ii) is reproduced correctly. For illustration, the development of the stress over the cross-sectional thickness for three selected levels of load is shown in Figure 15.

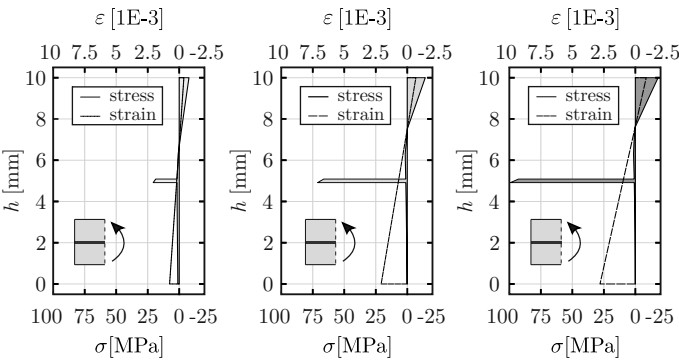

**Figure 15.** Development of strain and strain profiles of the carbon concrete beam with cross section (ii) according to Figure 14.

Let us note that, in case of a smeared cross-sectional model, the effect of fabric placement within the cross section cannot be captured. Considering the case of the studied girder with the fabric placed in the middle of the cross section, the smeared approach necessarily leads to an overestimation of the lever arm as can be recognized by comparing the stress profiles in a smeared and resolved cross section models depicted in Figure 10.

## 5. Finite Element Simulation of the Carbon Concrete Girder

Finite element model of the tested girder described in Section 1 has been decomposed into twelve zones to account for the different cross-sectional layups as shown in Figure 16. Sections 4–12 correspond to the reinforced sections displayed previously in the technical drawing in Figure 4. In addition to the reinforced sections, the finite element model also includes the non-reinforced boundary sections 1–3 positioned behind the supports.

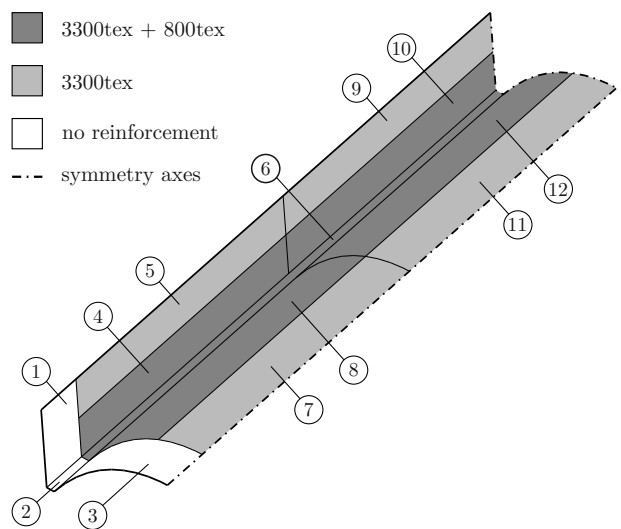

**Figure 16.** Cross sections of the girder with the corresponding layouts described in Tables 2 and 3.

The finite element discretization with the twelve sections is displayed in Figure 17. The finite element type was a bilinear quadrilateral layered shell element with three translational and two rotational degrees of freedom at each node. In the applied *ABAQUS* code, this element is referred to as a *conventional* shell element [31]. This type of element provides the possibility to define several layers within the cross section that can be associated with different material models and material parameters. This feature has been utilized to combine the microplane damage model for the concrete layers and the explicitly defined bilinear stress–strain curves for the fabric layers in a single cross section.

The symmetry of the shell was exploited to reduce the size of the simulated structure to a half of the girder. The load was introduced using displacement-control using stiff plates at positions indicated in Figure 17. To introduce the load from the plates to the shell consistently with the test setup, a frictionless contact was defined between the plates and the girder surface. The deflection of the girder was measured at the mid-span on the bottom side of the girder.

The described finite element model was used to study the effect of the cross section representation using four versions of cross section representation (i–iv) displayed in Figure 18. The simulated load–deflection curves corresponding to the four versions of the model are plotted in Figure 19 together with the test result. The obtained response curves are discussed in detail in the following sections.

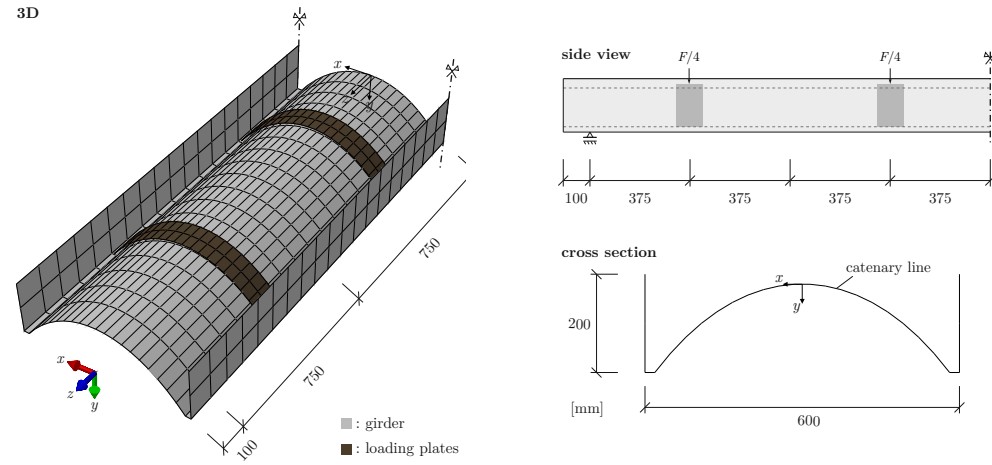

**Figure 17.** Finite element model of the lightweight girder: (**a**) finite element mesh and (**b**) dimensions of the girder cross section.

| model ╲ thickness | fabrics smeared in composite | fabrics resolved in layers |
|---|---|---|
| uniform consistent with tensile test | (i) | (iii) |
| nonuniform following the girder design | (ii) | (iv) |

**Figure 18.** Studied types of cross-sectional representations (i–iv).

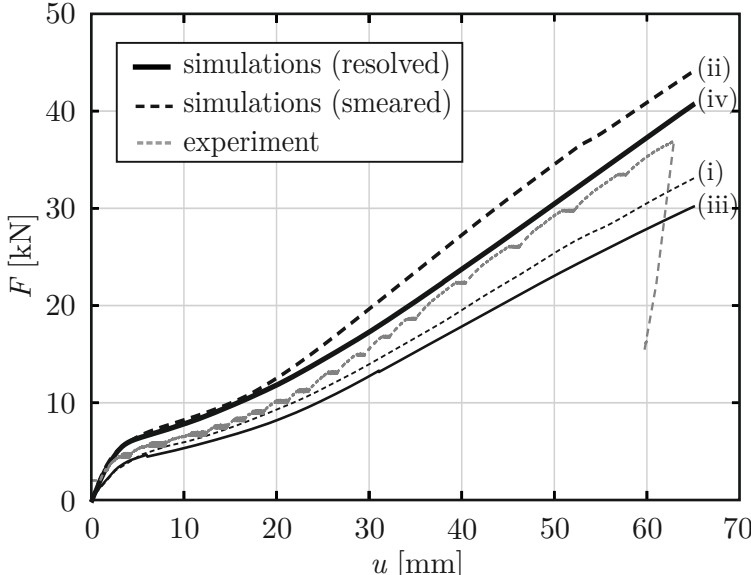

**Figure 19.** Simulated load–deflections curves in comparison with the test results.

### 5.1. Constant Thickness and Smeared Cross Section

To provide a reference example of the shell simulation with a simple non-scaled version of the cross-sectional parameters, a girder model with a constant thickness of 10 mm was used first. This cross-sectional model shown in Figure 18 (i) corresponded to the original cross section of the tensile test specimens used to identify the integrity functions.

Three calibrated integrity functions were used in this simulation to reflect the varying cross section within the shell. The zones 5, 7, 9, 11 reinforced with 3300 tex were associated with the strain-hardening behavior $\sigma_c^{3300}(\varepsilon_c)$ given in Figure 9d. The zones 4, 6, 8, 12 reinforced with hybrid fabric 3300 + 800 tex were associated with integrity functions calibrated using the stress–strain curve $\sigma_c^{3300+800}(\varepsilon_c)$ given in Figure 12b. The non-reinforced zones 1, 2, 3 were associated with a plain concrete strain-softening behavior $\bar{\sigma}_m(\varepsilon_c)$.

The obtained load–deflection curve (i) shown in Figure 19 has a lower final stiffness than the measured response of the girder test. The reason for the lower stiffness is the fact that the tested girder had additional layers of plain concrete (FCC) in zones 4–12 that were not included in this version of the model.

### 5.2. Varying Thickness and Smeared Cross Section

To evaluate the effect of additional concrete layers denoted as FCC in Figure 4, additional layers with strain-softening behavior $\bar{\sigma}_m(\varepsilon_c)$ have been added to the 10 mm thick shell model as indicated in Figure 18 (ii). The thickness parameters of the sections 1–12 are summarized in Table 2. The meaning of the parameters representing the added thickness in combination with the material model used for each layer is visualized in the cross-sectional layup depicted in Figure 20.

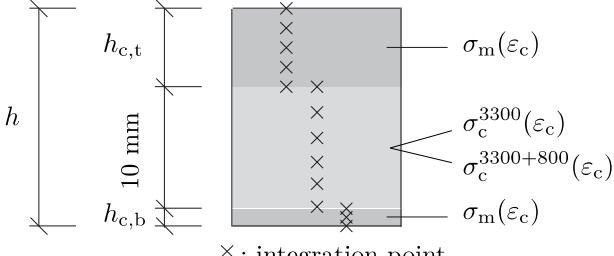

**Figure 20.** Cross-section with an effective strain-hardening layers either $\sigma_c^{3300}$ or $\sigma_c^{3300+800}$ representing the tested cross section and strain-softening layers $\sigma_m$ for add-on plain concrete layers.

**Table 2.** Cross section layups corresponding to the curve (ii) in Figure 16.

| Cross Section | $h_{c,b}$ [mm] | $h_{m,f}$ [mm] | $h_{c,t}$ [mm] | $h$ [mm] |
|:---:|:---:|:---:|:---:|:---:|
| 1 | - | - | - | 15.4 |
| 2 | - | - | - | 20.0 |
| 3 | - | - | - | 12.9 |
| 4 | 2.7 | 10 | 2.7 | 15.4 |
| 5 | 2.7 | 10 | 2.7 | 15.4 |
| 6 | 5.0 | 10 | 5.0 | 20.0 |
| 7 | 1.45 | 10 | 1.45 | 12.9 |
| 8 | 1.45 | 10 | 1.45 | 12.9 |
| 9 | 2.7 | 10 | 15.1 | 27.8 |
| 10 | 2.7 | 10 | 15.1 | 27.8 |
| 11 | 1.45 | 10 | 6.45 | 17.9 |
| 12 | 1.45 | 10 | 6.45 | 17.9 |

The load–deflection curve obtained using the model version (ii) is shown Figure 19. Compared to version (i), the additional concrete layers increase the structural stiffness. However, now the stiffness is even higher than the real stiffness of the tested girder.

Let us remark that, if there were no bending stresses in the girder, the smeared cross-sectional representation correctly reflecting the distribution of thickness and reinforcement ration in the shell surface should able to correctly predict the structural response of the girder. Thus, we can postulate that the amount of cross-sectional bending stresses in the shell was non-negligible and that the stiffness overestimation only reveals the deficit of the smeared model mentioned at the end of Section 4.5, i.e., that it is not able to reflect the effect of placement of the fabric within the cross section. In the studied girder, the fabrics were placed nearly in the middle of the cross section. Then, the smeared strain-hardening model leads to an overestimation of the lever arm. The situation is shown in Figure 10a, ascribing the same strain-hardening behavior to all integration points within the cross section. In such a case, the material points in the bottom region of the cross section cannot reflect the effect of cracks emerging in the tensile zone that would lead to a reduction of the lever arm.

### 5.3. Constant Thickness and Resolved Cross Section

To show the isolated effect of the lever arm on the load–deflection response, the girder was simulated with a constant thickness of 10 mm as done in version (i). The layered cross section with the fabrics placed in the middle is shown in Figure 18 (iii). As expected, the resulting load deflection curve (iii) of a thin girder with a constant thickness shown in Figure 19 exhibits a significantly lower stiffness in the post-cracking regime than the model (i) with the smeared cross section with the same thickness.

### 5.4. Varying Thickness and Resolved Cross Section

Finally, the resolved cross section model was combined with the additional layers of the plain-concrete $\bar{\sigma}_m(\varepsilon_c)$ similarly to the model version (ii). The parameters of the cross-sectional layup

corresponding to the 12 sections are specified in Table 3. The association of the layer thicknesses to the individual layers is depicted in Figure 21.

The calculated load–deflection curve (iv) reflects the stiffening effect of the additional plain concrete layers with respect to the curve (iii). At the same time, it shows the effect of an improved representation of the lever arm in the cross section compared to the model (ii) as documented by more realistic prediction of the real girder behavior as the level of force at the post-cracking stage and the final stiffness are predicted reasonably well.

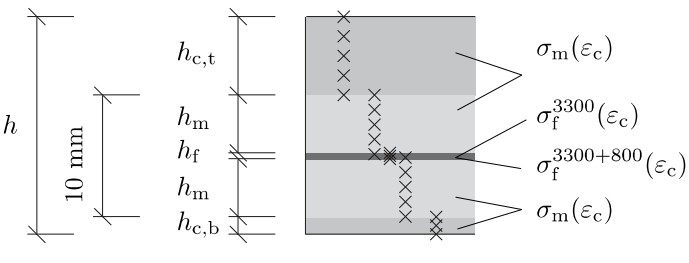

$\times$: integration point

**Figure 21.** Cross section with a resolved layer of fabrics and with strain-softening layers $\sigma_{\mathrm{f}}^{3300}$ or $\sigma_{\mathrm{f}}^{3300+800}$ and strain-softening layers $\sigma_{\mathrm{m}}$ for add-on plain concrete layers.

**Table 3.** Cross section layups corresponding to the curve (iv) in Figure 21.

| Cross Section | $h_{\mathrm{c,b}}$ [mm] | $h_{\mathrm{m}}$ [mm] | $h_{\mathrm{f}}$ [mm] | $h_{\mathrm{c,t}}$ [mm] | $h$ [mm] |
|---|---|---|---|---|---|
| 1 | - | - | - | - | 15.4 |
| 2 | - | - | - | - | 20.0 |
| 3 | - | - | - | - | 12.9 |
| 4 | 2.7 | 4.8835 | 0.233 | 2.7 | 15.4 |
| 5 | 2.7 | 4.9135 | 0.173 | 2.7 | 15.4 |
| 6 | 5.0 | 4.8835 | 0.233 | 5.0 | 20.0 |
| 7 | 1.45 | 4.9135 | 0.173 | 1.45 | 12.9 |
| 8 | 1.45 | 4.8835 | 0.233 | 1.45 | 12.9 |
| 9 | 2.7 | 4.9135 | 0.173 | 15.1 | 27.8 |
| 10 | 2.7 | 4.8835 | 0.233 | 15.1 | 27.8 |
| 11 | 1.45 | 4.9135 | 0.173 | 6.45 | 17.9 |
| 12 | 1.45 | 4.8835 | 0.233 | 6.45 | 17.9 |

Apparently, the force level in the stage of the multiple cracking is slightly overestimated. The possible reason is that the applied scaling rule does not account for the effect of the reinforcement ratio on the crack localization process in the matrix. An application of meso-scale models [17,18,32,33] with discrete crack representation that would additionally capture the interaction between damage localization in concrete and the stress transfer due to the finely distributed fiber and fabric reinforcement might help to analyze this effect in more detail. In addition, the interaction with the additional concrete layers made of a different material (FCC) has not been included in the model due to the lack of experimental data. A refinement of the scaling rules based on the meso-scale models and additional tests could significantly improve the prediction of the structural behavior in the service state.

Still, even in the present, pragmatic form, the applied modeling approach provides the possibility of deeper interpretation of the test results in terms of the visualized damage propagation through the thin shell. In Figure 22, the deformed shapes of the girder are plotted with the distribution of the

maximum damage values in each material point. The plot visualizes the damage variable $\omega$ defined within the microplane damage model as

$$\omega = 1 - \phi_{\min},\tag{15}$$

with $\phi_{\min}$ denoting the lowest microplane integrity level at a material point. The propagation of the damage through the shell is depicted in Figure 22 separately for the top and the bottom surfaces at three levels of loading corresponding to deflections $u_y = 20\,\text{mm}$, $u_y = 40\,\text{mm}$ and $u_y = 60\,\text{mm}$.

By comparing the bottom and top views on the shell at the three selected levels of load, the simulation results exhibit a difference between the damage values at the top and the bottom surface of the shell. Such a difference within a cross section is due to a significant amount of bending stresses within the cross section along the top middle section of the vault. This observation confirms the importance of the correct reflection of the changing lever arm during the multiple cracking phase of the test.

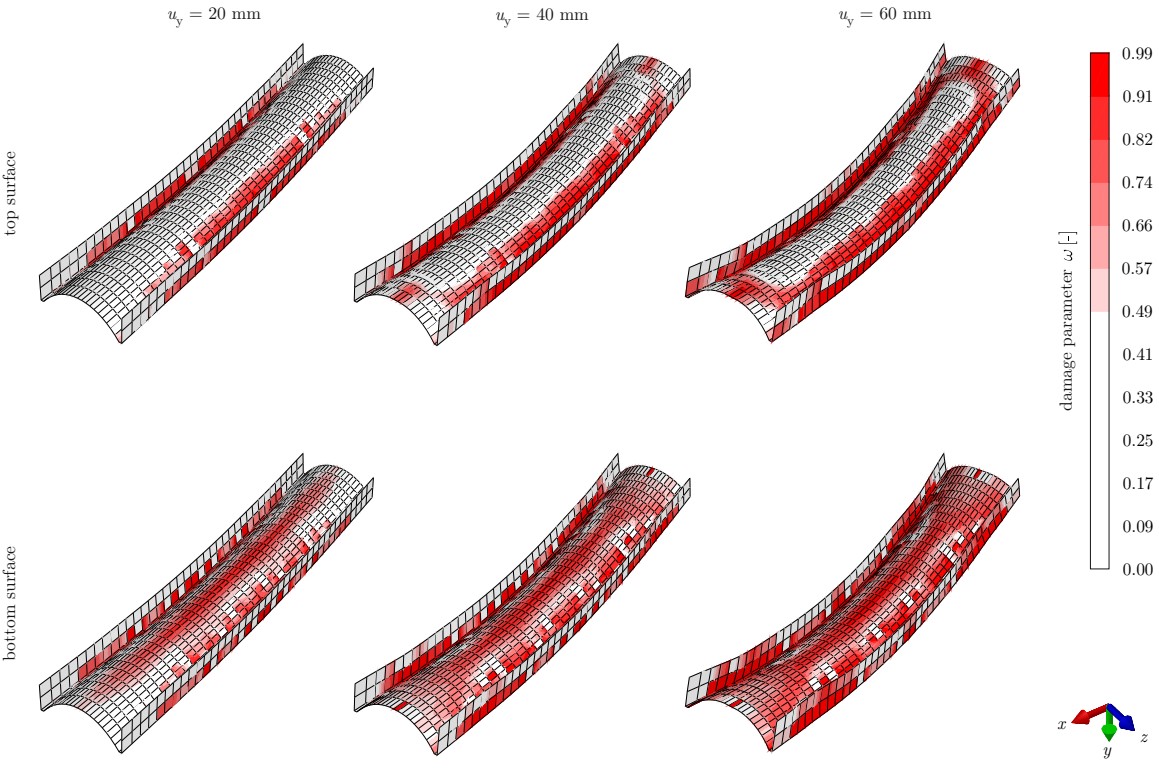

**Figure 22.** Damage propagation in the surface of the girder for top and bottom surfaces.

Furthermore, the damage distribution plots reveal the local effect of load transfer through the wooden plates. Even though the load was introduced via a frictionless contact model combined with the free displacement of the shell in the in-plane direction, local bending could be observed with the applied relatively coarse size of finite element. However, no significant influence on the overall structural behavior could be observed.

## 6. Conclusions

The strain-hardening behavior of textile-reinforced concrete has been modeled at the level of a cross section using both smeared and resolved representation. By decomposing the composite stress measured in the tensile test into the fractions associated with the fabric layer and with the matrix,

it was possible to define the mixture and scaling rules that allowed us to derive cross-sectional material parameters corresponding to changed layup and thickness.

The derived characteristics were applied in the simulation of the structural response of a girder element and compared with the experimentally obtained results. The systematically performed parametric studies confirmed the fact that the smeared representation of cross section is not directly applicable to shells with a nonuniform layup and variable thickness. Still, the smeared cross-sectional representation can be effectively used for layups with laminated fabrics with thin layers of cementitious matrix. This approach was used for non-penetrated fabrics uniformly distributed over the cross-sectional thickness in the applications presented e.g., in [34].

The resolved cross section model has shown a good agreement with the experimentally observed behavior. As such, it has the potential to provide valuable insight into the stress redistribution process during loading and can be used for further improvements to the geometrical design of the shell to achieve high structural quasi-ductility. Besides the validation of the resolved model using the girder test, the study was used to describe a general mixture and scaling procedure that can be applied to derive the wider range of cross-sectional characteristics from elementary tensile tests with a single layer fabric reinforcement.

The described simulation was primarily focused on the damage process inducing stress redistribution both along the thickness direction and in the in-plane direction of the shell surface. The ultimate failure was predicted reasonably well based on the strength measured in the tensile test specimens. Such a good prediction is due to the fact that critical cross section of the shell was primarily loaded in tension and the amount of bending at this particular location was negligible. As discussed in [13], the effective strength of the fabrics in cross sections exposed to bending can be significantly higher so that a refined ultimate-state criterion is needed that explicitly distinguishes the tensile strength of a cross section loaded in bending [35].

**Author Contributions:** Conceptualization,R.C.; methodology, R.C.; Software, E.S.; validation, E.S.; investigation, E.S. and T.S.-R.; data curation, T.S.-R.; writing—original draft preparation, R.C.; writing—review and editing, R.C. and E.S.; visualization, E.S. and T.S.-R.; supervision, R.C. and F.S.; project administration, F.S.

**Funding:** The theoretical part of this research was performed in Aachen and funded by the German Federal Ministry of Education and Research (BMBF, Grant No. 03ZZ0312C). The experimental part of the present work was conducted in Dresden and funded by the German Federal Ministry of Economic Affairs and Energy (BMWi, Grant No. KF2505611KI3).

**Conflicts of Interest:** The authors declare no conflict of interest.

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
