# Peer review of "Numerical Modeling of Non-Uniformly Reinforced Carbon Concrete Lightweight Ceiling Elements"

_applsci, doi:10.3390/app9112348_

Round 1

Reviewer 1 Report

Determination of material properties is very important in case of possibility to use them in numerical calculation. Taking above into account the subject of submitted paper as well as the proposed methodology allowing to determine the material properties of fibre in case of compression is very important.

The considered topic is very important and it is worth of publication but the paper should be improved. First of all it need claryffication, it is written in chaotic way, lots of repetitions sentens and informations, the Figures are placed far from the text describing it, some graphs are repeated as well (eg. Fig 7ab = Fig 9ab;  Fig 14cd = Fig 13ab.), there are Figures without proper references or lack of references in text (eg. Fig.9). The notation should be clearly explain.

1) page 4 line 105 - there is reference to Fig 2a but in paper there is only Fig 2 left, the same is with Fig. 2b
2) There are no information what are presented in Figs. 7a and 7b (axes and curves are the same on both graphs).
3) Fig. 7cd - the notation of vertical axis (fi) is not define (explained)
4) Page 8 line 213 "damage function" is not define

5) Explain how the reinforcement ratio  800+3300 have been calculated

6) page 12 line 281:the word "for" appears but what for ?

7) Fig. 21 what is relation between cases (i - iv) and different lines defined in legend and in comparison with cases (i-iv) presented in Fig.20 - the notation for cases denoted be (i) - (iv) should be claryfied - I noticed (maybe wrongly) that Authors used the same notation (i)-(iv) for different things.

I advice to read submitted paper carefully, correct it and resubmit, especially due to it is very interesting.

Author Response

Review #1

Determination of material properties is very important in case of possibility to use them in numerical calculation. Taking above into account the subject of submitted paper as well as the proposed methodology allowing to determine the material properties of fibre in case of compression is very important.

The paper primarily  focuses on the strain-hardening behavior in tension.  We only summarize the specific features of the behavior in the beginning of the paper. We hope that we could make the focus of the paper clearer by including a classification of modeling approaches to tensile response relevant for the scope of the paper in the introduction and added three-paragraph comment.

The considered topic is very important and it is worth of publication but the paper should be improved. First of all it need clarification, it is written in chaotic way, lots of repetitions sentens and informations, the Figures are placed far from the text describing it, some graphs are repeated as well (eg. Fig 7ab = Fig 9ab;  Fig 14cd = Fig 13ab.), there are Figures without proper references or lack of references in text (eg. Fig.9). The notation should be clearly explain.

This was indeed not good. We merged the mentioned figures. Originally we wanted to explicitly show the verification as well, but – the curve were just matching – so that we skipped it.

Fig 14cd was indeed redundant and removed. As a result, the numbers of the subsequent figures changed. Every Figure is now referenced at some point.

1) page 4 line 105 - there is reference to Fig 2a but in paper there is only Fig 2 left, the same is with Fig. 2b

Done

2) There are no information what are presented in Figs. 7a and 7b (axes and curves are the same on both graphs).

Done

3) Fig. 7cd - the notation of vertical axis (fi) is not define (explained)

Explanation of the integrity factor – now Fig. 9 in the caption added

4) Page 8 line 213 "damage function" is not defined

Damage function has been included in the description of the model in Sec 3 and the
paragraph explaining the issue of bending stiffness overestimation in the smeared model has been rewritten and simplified.

5) Explain how the reinforcement ratio  800+3300 have been calculated

equation added into Table 3.

6) page 12 line 281:the word "for" appears but what for ?

Corrected

7) Fig. 21 what is relation between cases (i - iv) and different lines defined in legend and in comparison with cases (i-iv) presented in Fig.20 - the notation for cases denoted be (i) - (iv) should be claryfied - I noticed (maybe wrongly) that Authors used the same notation (i)-(iv) for different things.

This was indeed wrong in the table and thus very confusing. We apologize for this.

I advice to read submitted paper carefully, correct it and resubmit, especially due to it is very interesting.

We feel a bit ashamed that the paper was not in a sound state for a final submission and thank the reviewer again for the work done on the review.

p { margin-bottom: 0.1in; line-height: 120%; }

Reviewer 2 Report

The paper discusses the structural analysis of thin-walled concrete shells through an approach to scale the material parameters for use in numerical simulations. A smeared modelling approach is compared to a resolved approach for the modelling of a carbon concrete girder. This is an important area of development  for layered TRCs. In general, the paper is well written, structured and the results are well presented.

The reviewer recommends to accept the paper after minor revisions/changes.

Remarks/suggestions:

Page 4, Figure 3: The figure shows the different layups as well as thicknesses of the shell at the different positions. If understood correctly, the left vertical part of the shell has a thickness of 15.4 mm, while at mid-span the thickness is 12.9 mm, however both regions seem to have the same textile reinforcement (3300 tex + 800 tex). Is it correct to assume the difference lies solely in the mortar thickness and both regions have the same amount of textile reinforcement? If yes, how does the fibre volume fraction of these regions relate to the fibre volume fraction reported in Table 1 or thus what is the reinforcement volume fraction in the different regions of the shell?

Page 4, line 104: The author states that the carbon fabrics are placed in the middle of the cross section. Since, as stated on page 8, lines 209 to 211: "In case that the reinforcement is positioned non-uniformly within the cross section, the model...fails to capture the bending behaviour of the shell". Why was the carbon layer therefore not placed in the bottom (tensile) region of the cross section in order to validate the correct lever arm representation of the resolved model and additionally, enhance the flexural mechanical response of the carbon concrete girder?

Page 4, line 106: Were any issues of bond (in terms of delamination) observed between the FCC and CRC layers?

Page 4, line 114: It is not clear for the reader what the 12 different cross-sectional layups composing the shell geometry are made of or how they differ from each other. Additionally, on Figure 18 it only looks like 3 different layups were used (3300 tex + 800 tex, 3300 tex, and no reinforcement)? It is proposed to clarify this by means of an additional drawing/table.

Page 4, line 118: How was the structural response of the girder measured? By means of LVDT's, strain gauges,...? It is proposed to clarify this in the text.

Page 6, line 180: The author states that "two series of tests with specimens reinforced with a single layer of fabrics were used, one with 3300 tex and one with 800 tex.". How many specimens were tested for each series and is there, to the author's belief, another reason than the slack phenomenon for the larger scatter of the experimental curves shown in Figure 7 (b) (3300 tex) compared to Figure 7 (a) (800 tex)? In general, it is expected that a larger fibre volume fraction would lead to less experimental scatter. 

Additionally to the previous comment: The calibration of the damage functions of the microplane-damage model has been extracted from the obtained stress-strain curves of these tensile tests on single-layer TRCs. However in these tensile tests, the textiles were placed centrally in a TRC cross section and the redistribution of stress concentrations over the specimen's thickness was unaffected by the presence of other textiles. What is the author's belief on the influence of the usage of these experimental curves for the prediction of layups containing two, closely stacked textile layers with respect to the possible stress redistribution/concentrations in the interface between the textile layers? It is suggested to add a short paragraph discussing this effect.

Page 7, line 186: A Young's modulus (E) of 28 . 10^5 MPa is reported, is this correct? A factor ten times lower is expected.

Page 15, line 327: The author states that conventional shell elements were employed for the numerical modelling of the carbon concrete girder. By default conventional shell elements have no thickness, therefore all boundary conditions and loading conditions have to be defined at the same position (over the height) of the shell (usually mid-plane). Even though thin, the real carbon concrete girder achieved thicknesses of almost 3 cm: what are the author's thoughts on the definition of boundary conditions and loads at the same geometrical plane and its influence on the obtained numerical simulation? Additionally, has the effect of employing a continuum shell (where the geometrical thickness can be represented) been investigated?

Page 16, line 357 and Figure 20/21: The numberings ((i), (ii), (iii) and (iv)) are not consistent between Figure 20, Figure 21 and the discussion presented in paragraphs 5.1, 5.2, 5.3 and 5.4. According to line 357 and Figure 21, number (ii) represents the model with varying thickness and smeared cross section while in Figure 20 (ii) corresponds to constant thickness and resolved model. This should be adapted.

Figures that have been employed in other works (e.g. Figure 5, 6, 8, 16) have to be referred to correctly. 

Corrections to language/form:

page 2, line 52: Add two dots ":" after distinguished in order to be consistent with the rest of the article.

page 2, line 53&54: Move the punctuation "." to after "elasticity modulus".

page 2, line 61: An "s" is missing in "These model".

page 2, line 72: It is suggested to use the terminology "plane stress" rather than "plain stress".

Page 5, line 145: It is proposed to add a comma (,) after "thermodynamically inconsistent".

Page 6, line 170+: Sometimes the wording "cross section" is used and sometimes "cross-section", it is suggested to be consistent throughout the entire article.

Page 6, line 177: An "s" is missing after "cross-sectional idealization".

Page 7, Figure 8: It is suggested to add a "+" rather than a "-" in the tensile region of strains, same comment in Figure 15.

Page 8, line 214: The word "only" is repeated.

Page 11, above Equation (5): "Is then reads" should be replaced by "then reads" or "is then".

Page 11, above Equation (7): The "s" at the end of "a hybrid fabrics" should be removed.

Page 12, one line under 280: An 's" is missing in "two different types of reinforcement layup".

Page 12, three lines under 280: Remove an "s" from "specimenss".

Page 12, line 281: The word "for" should be removed from this sentence.

Page 15, line 325: Correct the spelling of "disretization".

Page 18, line 387: It is suggested to remove "a" in "to analyze this effect in a more detail".

Author Response

Review #2

The paper discusses the structural analysis of thin-walled concrete shells through an approach to scale the material parameters for use in numerical simulations. A smeared modeling approach is compared to a resolved approach for the modelling of a carbon concrete girder. This is an important area of development  for layered TRCs. In general, the paper is well written, structured and the results are well presented.

The reviewer recommends to accept the paper after minor revisions/changes.

Remarks/suggestions:

Page 4, Figure 3: The figure shows the different layups as well as thicknesses of the shell at the different positions. If understood correctly, the left vertical part of the shell has a thickness of 15.4 mm, while at mid-span the thickness is 12.9 mm, however both regions seem to have the same textile reinforcement (3300 tex + 800 tex). Is it correct to assume the difference lies solely in the mortar thickness and both regions have the same amount of textile reinforcement? If yes, how does the fibre volume fraction of these regions relate to the fibre volume fraction reported in Table 1 or thus what is the reinforcement volume fraction in the different regions of the shell?

Table 1 only reviews the reinforcement ratios and characteristics related to the cross section with thickness d = 10 mm. These ratios are not related to the actual thickness variations in the shell. The simulation of the shell using the smeared cross-sectional model used in Figure 21 and 22 – variant (iii) was then augmented with additional plain concrete layers to represent the real thickness of the shell.

We haveaddedan additional explanation in Table 1 to clarify this.

Page 4, line 104: The author states that the carbon fabrics are placed in the middle of the cross section. Since, as stated on page 8, lines 209 to 211: "In case that the reinforcement is positioned non-uniformly within the cross section, the model...fails to capture the bending behaviour of the shell". Why was the carbon layer therefore not placed in the bottom (tensile) region of the cross section in order to validate the correct lever arm representation of the resolved model and additionally, enhance the flexural mechanical response of the carbon concrete girder?

The focus of the paper is not to improve the performance of the already tested shell but only to predict is behavior. It could certainly be improved as the reviewer suggests. But – next time ;-)

Page 4, line 106: Were any issues of bond (in terms of delamination) observed between the FCC and CRC layers?

There was no delamination observed between the layers. The multiple cracking of the matrix perpendicularly to the shell surface governs the inelastic behavior of the shell.

Page 4, line 114: It is not clear for the reader what the 12 different cross-sectional layups composing the shell geometry are made of or how they differ from each other. Additionally, on Figure 18 it only looks like 3 different layups were used (3300 tex + 800 tex, 3300 tex, and no reinforcement)? It is proposed to clarify this by means of an additional drawing/table.

Figures have been simplified and arranged in a better was so that we hope that the explanation is now clearer.

Page 4, line 118: How was the structural response of the girder measured? By means of LVDT's, strain gauges,...? It is proposed to clarify this in the text.

The deflection was measured in the mid-section of the shell at the bottom using LVDT’s.The information has been included.

Page 6, line 180: The author states that "two series of tests with specimens reinforced with a single layer of fabrics were used, one with 3300 tex and one with 800 tex.". How many specimens were tested for each series and is there, to the author's belief, another reason than the slack phenomenon for the larger scatter of the experimental curves shown in Figure 7 (b) (3300 tex) compared to Figure 7 (a) (800 tex)? In general, it is expected that a larger fibre volume fraction would lead to less experimental scatter. 

This is true in view of the fiber bundle theory regarding the stiffness and strength of filaments within the bundle. However, we cannot say at which level does the slack come into play. The sources of scatter come also at the level of fabrics and their waviness. Frankly speaking, we cannot answer the question as we do not have the data characterizing the applied fabrics. Slack of this type of fabrics is large and its scatter as well.

Additionally to the previous comment: The calibration of the damage functions of the microplane-damage model has been extracted from the obtained stress-strain curves of these tensile tests on single-layer TRCs. However in these tensile tests, the textiles were placed centrally in a TRC cross section and the redistribution of stress concentrations over the specimen's thickness was unaffected by the presence of other textiles. What is the author's belief on the influence of the usage of these experimental curves for the prediction of layups containing two, closely stacked textile layers with respect to the possible stress redistribution/concentrations in the interface between the textile layers? It is suggested to add a short paragraph discussing this effect.

This is indeed an important point. The direct stacking of fabrics brings about the danger of delamination due to the low amount of matrix within the fabric mesh connecting the concrete layers. Still, within the tested girder, this effect has not been obverved. There is a short comment on this issue at the bottom of page 11.

Page 7, line 186: A Young's modulus (E) of 28 . 10^5 MPa is reported, is this correct? A factor ten times lower is expected.

No, sillymistake – corrected, apologies ...

Page 15, line 327: The author states that conventional shell elements were employed for the numerical modelling of the carbon concrete girder. By default conventional shell elements have no thickness, therefore all boundary conditions and loading conditions have to be defined at the same position (over the height) of the shell (usually mid-plane). Even though thin, the real carbon concrete girder achieved thicknesses of almost 3 cm: what are the author's thoughts on the definition of boundary conditions and loads at the same geometrical plane and its influence on the obtained numerical simulation? Additionally, has the effect of employing a continuum shell (where the geometrical thickness can be represented) been investigated?

The conventional shell was used with a layered representation of the cross section as it is provided in Abaqus. The vertical displacement at the position of supports was fixed. Since the principle tensile stresses were oriented in the longitudinal direction of the girder so that the problem that the reviewer mentions – of a local 3D stress state at the supports did not occur. This might be indeed a problem if local bending moments were induced in the support region in the direction perpendicular to the girder.

Page 16, line 357 and Figure 20/21: The numberings ((i), (ii), (iii) and (iv)) are not consistent between Figure 20, Figure 21 and the discussion presented in paragraphs 5.1, 5.2, 5.3 and 5.4. According to line 357 and Figure 21, number (ii) represents the model with varying thickness and smeared cross section while in Figure 20 (ii) corresponds to constant thickness and resolved model. This should be adapted.

This was an error – apologies !!

Figures that have been employed in other works (e.g. Figure 5, 6, 8, 16) have to be referred to correctly. 

Reference to the paper where they’ve been shown.

Corrections to language/form:

Aplogies for so many typos, many thanks for this extensive list, we made the correction and proof reading

page 2, line 52: Add two dots ":" after distinguished in order to be consistent with the rest of the article.

page 2, line 53&54: Move the punctuation "." to after "elasticity modulus".

page 2, line 61: An "s" is missing in "These model".

page 2, line 72: It is suggested to use the terminology "plane stress" rather than "plain stress".

Page 5, line 145: It is proposed to add a comma (,) after "thermodynamically inconsistent".

Page 6, line 170+: Sometimes the wording "cross section" is used and sometimes "cross-section", it is suggested to be consistent throughout the entire article.

Page 6, line 177: An "s" is missing after "cross-sectional idealization".

Page 7, Figure 8: It is suggested to add a "+" rather than a "-" in the tensile region of strains, same comment in Figure 15.

Page 8, line 214: The word "only" is repeated.

Page 11, above Equation (5): "Is then reads" should be replaced by "then reads" or "is then".

Page 11, above Equation (7): The "s" at the end of "a hybrid fabrics" should be removed.

Page 12, one line under 280: An 's" is missing in "two different types of reinforcement layup".

Page 12, three lines under 280: Remove an "s" from "specimenss".

Page 12, line 281: The word "for" should be removed from this sentence.

Page 15, line 325: Correct the spelling of "disretization".

Page 18, line 387: It is suggested to remove "a" in "to analyze this effect in a more detail".

Reviewer 3 Report

General Comments

The article treats an interesting issue related to the numerical modeling of thin shell elements, composed of reinforced concrete with textile carbon fabrics. The authors have conducted tests on a lightweight ceiling prototype and advanced numerical simulations based on the ABAQUS platform. The methodology here conducted is relevant and scientifically sound.

The manuscript fulfills the scope of the journal. In this regard, the paper deserves the attention of the editorial board. However, for the sake of clarity, the submitted manuscript may be considered for publication also taking account for the above suggestions, and professional English proofreading.

Specific Comments

·         In section 3, the authors refer to the anisotropic damage model. However, in the text, many aspects are not very clear or properly highlighted. It results that the focus is on the heterogeneity of the cross section (including the variability of the fabric position in the thickness) than to the anisotropy! The authors are asked to clarify this aspect and in addition, enriching the numerical model with more explanation and data for the sake of the clarity.

·         Figure 18, and the respective tables 2 and 3 are very crucial for their research. It is not very clear the choice of the values. For instance, why section 5 and 9 should defer? The variability of the properties could be interesting to be correlated to a practical case.

·         The use of the term “smeared cross-section” may not be an adequate one. In this regard, and not only, but the authors should also further explain their used approach for the numerical model. Is it a single material or with multiple layers where different material properties may be assigned?

·         The results of figure 24 are not compared with test results, thenceforth are not either comparable or reliable.

·         The authors are asked to highlight better the practical usability of their approach with pros and cons. As far as it is may be worthy, it would be beneficial to represent other modeling approaches of what concerns the micro modeling. 

·         Some remarks on the numerical approach should be done. The approach, as it is here presented is slightly confusing and a lack of references is notable. An extensive self-citation, compared to other references, is used by the authors, which is not really appreciated. Such an aspect should be reviewed.

Author Response

Review #3

General Comments

The article treats an interesting issue related to the numerical modeling of thin shell elements, composed of reinforced concrete with textile carbon fabrics. The authors have conducted tests on a lightweight ceiling prototype and advanced numerical simulations based on the ABAQUS platform. The methodology here conducted is relevant and scientifically sound.

The manuscript fulfills the scope of the journal. In this regard, the paper deserves the attention of the editorial board. However, for the sake of clarity, the submitted manuscript may be considered for publication also taking account for the above suggestions, and professional English proofreading.

Specific Comments

·         In section 3, the authors refer to the anisotropic damage model. However, in the text, many aspects are not very clear or properly highlighted. It results that the focus is on the heterogeneity of the cross section (including the variability of the fabric position in the thickness) than to the anisotropy! The authors are asked to clarify this aspect and in addition, enriching the numerical model with more explanation and data for the sake of the clarity.

We are sorry if the distinction of the cross sectional layup (heterogeneity as the reviewer calls it) and the damage induced anisotropy was not clear. We hope that the present version of the paper explain the issues better. We rewrote the section 3 and hope that it is clearer now. The model itself is not the primary focus of the paper so that at some point we refer to the original paper describing it in detail. We also write that other inelastic strain-hardening and stress-softening modes (plasticity based) can be applied in combination with the specified decomposition → scaling → mixture formulas.

·         Figure 18, and the respective tables 2 and 3 are very crucial for their research. It is not very clear the choice of the values. For instance, why section 5 and 9 should defer? The variability of the properties could be interesting to be correlated to a practical case.

Regions 5 and 9 have different layups – thickness – as gigen in tables 2 and 3 as visualized in Fig. 3 inoriginal submission, now Fig 4,, so that they must be distinguished in the model. Regarding the practical applicability: The last sentence is not clear to us. What is meant with variability? The last version of the model (iv) corresponds to the real/ractical design of the girder.

·         The use of the term “smeared cross-section” may not be an adequate one. In this regard, and not only, but the authors should also further explain their used approach for the numerical model. Is it a single material or with multiple layers where different material properties may be assigned?

We hope that the newly added Fig. 1 can better clarify the terminology and visualize the approach used in the smeared and resolved representation of the stress-strain behavior.

The same material behavior is ascribed to any material point within the cross section. Alternative terms like “uniform, homogeneous or effective” are might be possible but in our opinion they do not emphasize the deliberate choice of the modeling approach – to smear the behavior over a region of the cross section. Therefore we consider this term adequate.

·         The results of figure 24 are not compared with test results, thenceforth are not either comparable or reliable.

It is not clear what the reviewer suggests. The comparison between the numerical prediction and between the test is done in a usual way in terms of the measured load-deflection curve in Fig. 21 (numbering in original submission). Fig. 24 complements the information with the distribution of the damage state variable indicating which regions of the shell entered the inelastic regime of material behavior. What would the reviewer want to compare? Monitoring of the fine crack pattern in the test? That’s would be hugely expensive (fiber sonsor system?). This requirement does not seem to be relevant.

·         The authors are asked to highlight better the practical usability of their approach with pros and cons. As far as it is may be worthy, it would be beneficial to represent other modeling approaches of what concerns the micro modeling. 

We have added additional remarks and references to further model concept. There are now 35 references that are indeed relevant for the focus of the paper. We mention the multi-scale types of modeling explicitly representing individual cracks in the introduction. Fig 1 tries to indicate the basic principles of the different approaches and their relations. We add the remark that application of approaches with fine resolution of the material structure and of individual cracks in the model of a shell at hand is practically infeasible.

The paper is focused on layered shell representation of TRC shells and not primarily on the particular numerical modeling approach. In particular, decomposition, scaling and mixture rules for a strain hardening cross section are the key issues. These concepts are not linked with a a particular material model and can be applied in combination with any layered shell finite element and material model, no matter if damage or plasticity based. The anisotropic damage model serves as an example and is not part of the key concept.

Practicable usage is provided by showing that the model simulates can realistically predict the response of the tested girder.

·         Some remarks on the numerical approach should be done. The approach, as it is here presented is slightly confusing and a lack of references is notable. An extensive self-citation, compared to other references, is used by the authors, which is not really appreciated. Such an aspect should be reviewed.

With the numerical approach, the reviewer probably means the material model. As said above, the material is just has a role of a running horse to demonstrate the concept and to validate the approach. Therefore, it does not deserve much space in the present paper and we delegate the description to our detailed description elsewhere. Related citations setting the approach into the microplane framework are provided as well. In this view we consider the to related work adequate. We added further references to broader range of assessment, design rules and to models focused on TRC shells. However, in our opinion, further additions would just distract from the key message provided by the paper.

p { margin-bottom: 0.1in; line-height: 120%; }

Round 2

Reviewer 1 Report

Authors after the first review correct the paper especially the quality of presentation - in present version it could be published

Reviewer 2 Report

Thank you for thoroughly addressing the different questions/remarks. The paper can be accepted in its revised form.